# Stress among caregivers of autistic children: Conceptual analysis and verification using two qualitative datasets

**Stephen James Gentles**[1]*, **Janet McLaughlin**[1], **Margaret A. Schneider**[2]

**1** Health Studies, Wilfrid Laurier University, Waterloo, Ontario, Canada, **2** Department of Kinesiology & Physical Education, Wilfrid Laurier University, Waterloo, Ontario, Canada

* stevegentles@gmail.com

## Abstract

In this two-part study, we first present the results of a sub-analysis of empirical data from a large grounded theory study of caregivers' (parents) navigating autism-related care. The purpose of this analysis was to develop a conceptual overview of stress and crisis. We then describe the results and feasibility of using a framework analysis approach to verify and extend this conceptual analysis using qualitative survey data from a comparable population. Finally, we compare the conceptual findings to existing stress theory. While the grounded theory analysis was not aimed at producing a full theory of stress, multiple key elements of the resulting conceptual overview are consistent with prior stress theory. A potentially novel contribution is the conceptualization of social-psychological stress as an evolving process metaphorically analogous to a physiological model of stress that accurately fitted caregivers' experience. Specifically, it accounts for early empowering consequences of stress in terms of caregivers' motivation and capacity for action, the progressive destructive consequences in terms of its effects on caregivers' emotional and even physical well-being, and the evolving and nonlinear process of stress over the life course. The definition for crisis, meanwhile, acknowledges that different systems can be in crisis from the caregiver's perspective, and that it can be triggered by progressive buildups of stress and not just acute major triggering events. The insights from this analysis have implications for improving support professionals' sensitivity to the empirical caregiver-perspective realities of stress at a conceptual level, and for improving assessment of crisis specifically in this population. The framework analysis exercise demonstrated some utility of the qualitative survey data for verifying and extending this theoretical analysis, despite the limitations compared to in-depth interview data discussed. This has implications for improving the utilization of qualitative data often collected in survey research.

## Introduction

Autism is a complex, lifelong, neurodevelopmental condition estimated to be diagnosed in 1 in 36 children [1]. Its prevalence is not considered to be substantively affected by geography,

**Data Availability Statement:** Data cannot be shared publicly because of ethical privacy and confidentiality requirements related to the grounded theory study's qualitative interview data

(as enforced by the Hamilton Integrated Research Ethics Board (HiREB)), and the survey study's qualitative survey data (as enforced by the Wilfrid Laurier University Research Ethics Board). The enforcement by both REBs is governed by Canada's Tri-Council Policy Statement: Ethical Conduct for Research Involving Humans – TCPS 2 (https://ethics.gc.ca/eng/tcps2-eptc2_2022_chapter5-chapitre5.html#2), and corresponds to the required legal and ethical assurances provided to participants in the consent process about how their privacy and the confidentiality of their data would be protected. Grounded theory or survey data are available for researchers who meet the criteria for access to such confidential data after first contacting either the HiREB (fspence@mcmaster.ca) or Wilfrid Laurier University REB (REB@wlu.ca), respectively.

**Funding:** "SJG received a grant from Autism Ontario to support development of this manuscript (grant number N/A). SJG completed the survey data analysis during a postdoctoral fellowship at Wilfrid Laurier University, funded by a gift to JM and the Laurier Autism Research Consortium from the Dare Family, in honor of the late Carl M. Dare (grant number N/A). SJG completed the primary grounded theory study during his doctorate at McMaster University, with the support of two Strategic Training Initiative in Health Research fellowships funded by the Canadian Institutes of Health Research—an Autism Research Training Fellowship jointly funded by the Sinneave Family Foundation, and a Knowledge Translation Canada Fellowship (grant: CIHR 2777601301). JM received an internal grant from Wilfrid Laurier University to support the survey (grant number N/A). The funders had no role in study design, data collection and analysis, decision to publish, or preparation of the manuscript."

**Competing interests:** The authors have declared that no competing interests exist.

race, or socioeconomic factors [2]. Diagnostically, it is defined by differences in social communication and interaction, and restricted, repetitive patterns of behavior, interests, or activities [3]. As a holistic condition, autism is also increasingly recognized to be characterized by different strengths [4]. Its complexity is defined by pronounced variations in behaviours and disability in different areas [5], and by varying developmental trajectories [6].

Due to the high level of child-to-child variation and numerous possible autism-related concerns, parental caregivers (hereafter caregivers) are the de facto experts on their child's unique needs. Consequently, caregivers are also the main drivers of care (including wide-ranging interventions, services, and supports). Autism-related care, however, is also complex—spanning health, education, social services, and other support systems. Moreover, for each caregiver-defined autism-related concern, the caregiver is commonly responsible for identifying and accessing the appropriate care, which often involves multiple interventions or accommodations per concern. In an earlier qualitative study, Gentles and colleagues [7] labelled this process *navigating care* (or *intervention*). Caregivers' navigating process is long-term, evolves over time, and involves many care-specific tasks, including researching, decision-making, gaining access (e.g., completing forms), financing, budgeting, planning (e.g., meetings), coordinating and traveling to appointments, managing financial and human resources, negotiating obstacles (e.g., barriers in the education system), engaging in personal development to acquire skills, participating in treatment implementation or modification, monitoring implementation, and evaluating effectiveness. Combined, these tasks require considerable work, rapid development of expertise (knowledge, skills), and expendable personal resources (time, energy and financial resources) from the caregiver. Navigating care therefore represents a major burden and source of stress for caregivers [7].

While navigating autism-related care is a significant stressor for most caregivers, greater research attention has typically been paid to understanding stress attributable to caregivers' *parental caregiving role*. Several studies, for example, have established the importance of child behaviors requiring parental management as a key source of maternal stress [8–11]. However, understanding more about the *varied sources* of stress experienced by caregivers of autistic children—those attributable to both their *parenting* and their *care-navigation roles*—is warranted because of the especially high levels and many known consequences of stress in this population.

Numerous studies have characterized the disproportionately high levels, and social consequences of stress among caregivers of autistic children—including depression [12–16], anxiety [12, 17], combined psychological distress [8, 10, 18, 19] reduced parenting self-efficacy [20, 21], impaired family functioning [10, 22], marital problems and divorce [19, 23, 24], and poorer physical health [19, 21, 25, 26]. In prior qualitative work, Gentles and colleagues [7] described these as *secondary* autism-related concerns, because they affect health or well-being at the caregiver or family level, and warrant dedicated intervention or care to address. Some research has also shown how caregiver stress and its consequences can feed back to impact child well-being and family functioning [20, 27, 28], and impede caregivers' abilities to participate in interventions directed at helping the child [29] or parental self-efficacy [21].

A potential reason why caregiver stress attributable to care navigation has been less studied than parenting stress is the lack of measures of the impacts of care navigation. In addition to being a primary cause of stress, however, care navigation has potentially measurable impacts on personal and family time, energy, and financial resources. Such navigation-related *expendable resource loss* represents another source of stress, through the additional stressful problems it causes (i.e., secondary concerns) including financial insecurity, chronic sleep deprivation, physical and emotional exhaustion, physical health impacts due to lack of self-care, family dysfunction and breakdown, and mental health strains including the emotional effects of guilt for

not doing enough. These are additional to the more intrinsic sources of stress related to navigating care, including caregivers' sense of urgency, and the inevitable barriers and conflicts they encounter in this process [7]. Tracing the distal problems resulting from caregivers' multi-factorial stress back to their process of navigating care through a dedicated qualitative analysis has the potential to improve understanding the mechanisms underlying stress-related concerns, and the theoretical foundations both for more relevant measurement and intervention to appropriately address them.

Crisis from the caregiver perspective is another important stress-related phenomenon, which in the case of autism can occur at multiple points across a child's lifespan, and is often associated with utilization of emergency services. Reflecting broadening awareness of the high prevalence of crisis among caregivers and families of autistic children, efforts to study it are increasing. The autism-specific Brief Family Distress Scale (BFDS) is a single-item measure of caregivers' perception of family distress that asks caregivers to identify proximity to crisis, and has been positively associated with stressors such as aggressive behavior and negative life events [30]. High baseline BFDS ratings have been associated with subsequent emergency department visits for any reason [31]. While the BFDS is based on an operational definition for crisis, it was not developed with specific reference to caregiver experience in the context of autism. There is no published caregiver-perspective definition of crisis, nor accounts of the relationship between crisis and caregivers' care navigation process to our knowledge—both of which may support future measurement and study of crisis in this group.

In this two-part paper we first present the primary definitions and conceptual overview of stress and crisis that were developed from empirical data in a large grounded theory study of caregivers' process of navigating autism-related care. Second, we describe the results and feasibility of subsequently using a *framework analysis* approach [32–34] to verify and extend this conceptual analysis using stress-relevant qualitative data from a later study: a survey study of caregivers of autistic children from the same geographic region (Ontario, Canada). Finally, we compare the conceptual findings to existing stress theory to highlight consistencies and possible new insights.

## Methods

### Primary conceptual analysis of grounded theory data

The primary definitions and conceptual analysis of stress and crisis presented in the first part of this paper were developed from a grounded theory study whose goal was to develop a substantive theory of how caregivers navigate autism-related care from the time they first perceive developmental concerns in their child [35]. Importantly, that study's aim was not to develop a full theory of stress, and thus our conceptual analysis does not provide a comprehensive mechanistic account. The grounded theory methods and rigor used for this study [36–39] have been described previously [7, 40]. The primary data comprised 45 verbatim transcripts of 90-minute intensive interviews with 32 caregivers (four completing two interviews) and 9 professionals selected for extensive experience supporting caregivers—representing a large participant and interview data sample for grounded theory [41].

As described previously [7, 40], caregiver backgrounds varied widely in terms of rural (22%) and urban (78%) status, child ages (range 2.5–18 years), child's autism severity level, number of autistic children (13% with multiple, maximum five children), ethnocultural backgrounds (34% non-Canadian), education (from high school diploma [9%] to graduate degree [16%]), and time spent navigating care at time of interview (<1 year to 8+ years). Of all aspects of caregivers' backgrounds, time navigating care was the most relevant metric from life course and process-oriented perspectives for analyzing differences in stress as it related to the evolving

process of navigating care. Importantly, the concept and terminology for reporting children's "autism severity level" (mild, moderate and severe) below is outdated due to the version of the diagnostic manual, the DSM-IV-TR, that was in use when most children in this study received their autism diagnosis; it provides only an imperfect estimate of child support needs.

Using a flexible interview guide, iteratively revised to address emerging theoretical foci according to theoretical sampling, participants were asked about experiences and actions navigating autism-related care over the child's life course. Participants all described a prominent role for stress, regardless of background, and the caregiver and professional interview guides were adjusted early in the study to include questions about stress and crisis (see S1 Appendix). Interviews thus produced a substantial body of qualitative data on the topic, including antecedents and consequences of stress, which enabled the current conceptual analysis.

All interview data were open-coded and incorporated into the final analysis. Analysis included constant comparison throughout coding and category development, analytic memo writing, developing abstract conceptual diagrams, and final integrative writing of the theory. Firsthand person-centered perspectives were prioritized both by the grounded theory methods, and the social theory of symbolic interactionism [42] used as an explicit analytic framework. Verification of the evolving categories proceeded with successive rounds of data collection informed by the ongoing analysis, per theoretical sampling. Notably, the typology of stress elaborated below was tested in a dedicated internal verification step by selectively coding roughly 50 previously incompletely-coded but relevant interview segments (open coded early in the grounded theory analysis prior to analytic development of the typology). Analysis and data management were supported by software (NVivo; QSR).

### Role of a priori knowledge

Given the existing theory on stress, an important methodological consideration for this study is when and how possibly relevant prior theory was used in the grounded theory analyses regarding stress and crisis. Grounded theory methods require that the substantive theory produced be developed firstly from the empiric-level study data. Furthermore, while any pre-existing knowledge the analyst arrives with is accepted as an inevitable source of *theoretical sensitivity* that informs the resulting analysis and theory construction, researchers are not expected and even discouraged from seeking additional knowledge of relevant theory until analysis is complete [36, 37, 43]. Thus, the lead analyst did not expand their knowledge of stress theory until study completion. It remains important for grounded theory analysts to report reflexively on their prior theoretical knowledge, and how it influenced their analysis [44]. In this case, prior knowledge of formal stress theory was minimal and limited to Seyle's General Adaptation Syndrome [45]. This physiological stress theory was applied as a metaphor for organizing the time-evolving aspects of caregivers' stress process. Its coherence (consistency) with the empirical data was verified throughout the analysis per grounded theory requirements. The constructs presented in this paper were not explicitly informed by other well-known stress theories, although they have been related to other such theories in the discussion.

### Crisis definition

The definition for crisis, meanwhile, was influenced by prior knowledge of the definition used in developing the BFDS (described above for measuring proximity to crisis among caregivers of autistic children) [30]:

> "*An acute disruption of psychological homeostasis in which one's usual coping mechanisms fail and there exists evidence of distress and functional impairment. The subjective reaction to*

*a stressful life experience that comprises the individual's stability and ability to cope or function. The main cause of a crisis is an intensely stressful, traumatic, or hazardous event...*"
[46]

This definition was adapted and revised through constant comparative analysis to fit caregivers' responses to questions about what crisis meant to them, and to iterative interpretation of coded examples of crisis in the data.

## Conceptual verification and extension using survey data

In the subsequent analysis exercise to verify and extend the primary conceptual analysis on stress, we coded stress-relevant qualitative data from a separate survey study, whose methods were described previously [19]. The survey's purpose was to explore the impacts of local autism policy, and barriers and facilitators to accessing private and publicly funded services. It featured 179 closed and open-ended questions, completed online by 654 Ontario caregivers of autistic children three years after the grounded theory data collection—representing a geographically similar population in which both samples shared the same autism policy and service landscape. Nine open-ended questions containing stress-relevant data were coded (see S2 Appendix). Unlike questions in the grounded theory interview guide, these asked less directly about stress or crisis, and several questions addressed service-related aspects expected to be associated with stress. Codable responses ranged from several words to several lines of participant-typed text.

## Framework analysis approach

The *framework analysis* approach followed to code survey responses has been used to analyze primary data in qualitative research in similar situations [32–34]. Framework analysis has been further adapted as an analytic method for synthesizing published qualitative study findings [47, 48], resulting in the related method named *best-fit framework synthesis* [49–52]. In both methods, new qualitative data are analyzed using a pre-existing theoretical framework, chosen for its relevance to the topic under study, and used as an initial coding framework. Themes, categories, or concepts from the pre-existing framework—in this case, the stress-related typology and categories from the grounded theory study—are thus used as an *a priori* coding structure. This framework functioned to organize the data to be coded, but was left open to refinement and extension as data from the new source (the open survey responses) were compared but found to represent novel ideas unaccounted for by the framework, allowing inductive generation of new open codes. Open codes from the survey data were considered new if they did not have a similar counterpart within the grounded theory coding structure.

Results of the survey data analysis do not feature example quotes because of ethics requirements. Thus, results of the survey data coding and framework analysis exercise is presented in aggregate summary form only. We present these results as answers to two questions derived from the aims of framework analysis: 1) What can be said about how the survey data replicates the stress-related coding structure from the grounded theory study? and, 2) How does the survey data extend the stress-related coding structure of the grounded theory study? In the discussion, we consider the methodological learnings and utility of the framework analysis exercise using survey data.

## Ethics statement

The grounded theory study described in this report received ethics approval from the Hamilton Integrated Research Ethics Board, and all participants provided prior written and ongoing

verbal consent for interviews (recruitment: 2011-October-24 to 2012-November-26). The survey study described in this report received ethics approval from the Wilfrid Laurier University Research Ethics Board, and all participants provided electronic written consent (recruitment: 2018-May-03 to 2018-July-22).

## Results

We first present the definitions of stress and crisis, followed by a typology of conceptual categories of caregiver stress (its main causes or sources, and associated consequences), which were developed and iteratively verified from caregiver accounts in the grounded theory study.

### Definitions of stress and crisis

*Stress* was defined as a caregiver's subjective emotional and physiological experience (which caregivers described in emotional and physical terms) and response to a triggering thing or event that they perceive as a threat to personal well-being (including that of a child) or continued ability to function. The *physiological idea of stress* from Seyle's General Adaptation Syndrome [45]—where stress is understood to mobilize increased cognitive and physical energy for responding to threatening situations in both the short term (alarm, or fight-or-flight, reaction) and the medium term (resistance), but leads to negative consequences when stress is overly prolonged (exhaustion)—was a useful, empirically coherent analogy for interpreting and explaining caregiver accounts of stress in its varying forms. Specifically, it helped account for the earlier empowering consequences of stress in terms of caregivers' motivation and capacity for action, the destructive consequences in terms of its effects on caregivers' emotional and even physical well-being, and the evolving nature, or process, of stress. This analogy fit empirically with the timeline of the different consequences of stress manifested in most caregivers' trajectories of navigating care—with effects on motivation and capacity for action happening earlier in the trajectory, and negative consequences happening throughout but changing in nature, with exhaustion manifesting over the longer term. The rough timing of these differing sources and consequences of stress is depicted in the model below (Fig 1), with time in caregivers' care navigation trajectory represented from left to right. The model of four main categories (causes) of stress derived from the grounded theory is elaborated (with examples from the data) after introducing the definition for crisis, below.

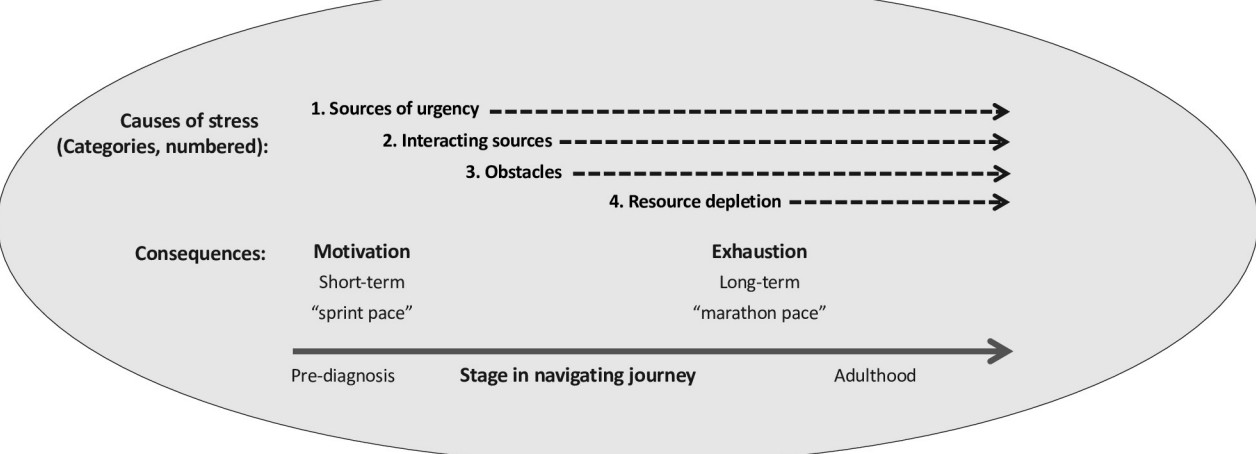

**Fig 1. Model of four main categories (causes) of caregiver stress, and their consequences for caregiver action and emotional and physical well-being over time, derived from the grounded theory analysis.**

*Crisis* was defined as an event occurring when caregiver stress (due to parenting, navigating care, etc.) reaches or passes a threshold level that disrupts homeostasis of a caregiver-related functioning system (listed below), resulting in a sudden or progressive loss of function, and sometimes the sense that a catastrophic failure is imminent. Crisis compels the caregiver to respond by seeking either resolution to the problem, or some other change from the status quo to restore balance. Failure of the system can occur but is generally rare. Two important qualifications captured caregivers' varying experiences and accounts of crisis:

1. Different functioning systems can be in crisis including the caregiver's self (comprising emotional/psychological, physical, or financial aspects), the family, spousal unit, child, or caregiver-child unit. Thus, for any example of crisis it is important to identify the system that crisis centers around.

2. The triggering cause of crisis can include not only a single acute stressful event, but also (more commonly) a progressive buildup of stress where the final threshold-breaking event can seem relatively benign (like the straw that breaks the camel's back). One caregiver described crisis this way:

*"I would say the crisis then for us was several peaks like that where they were just too close...I don't think there's ever been a time where it's been one specific [trigger of] crisis. It's always been too many things. When it rains, it pours." (mother from non-metropolitan city, one 17-year-old autistic son, originally diagnosed with what the parent recalled as 'severe autism')*

In most cases in the data, caregivers clearly and explicitly attributed crisis to a buildup of stress-related precursors. In the following caregiver's example, the buildup of stressors led to multiple systems threatened by crisis, and ultimately her response to restore balance: Dilin (pseudonym; divorced mother of one 6.5-year-old autistic son, originally diagnosed as 'severe' per parent's recollection, and two non-autistic siblings) described stress from exhaustion due to prolonged expendable resource loss (i.e., of time, finances, and physical and emotional reserves for coping), which became compounded by a series of additional stressors. She suffered from both generalized anxiety disorder triggered by navigating intervention, and depression due to overextending herself that led her to shut the world out. Additionally, she described how a combination of disagreement with her husband about approaches to intervention and her neglect of their relationship after dedicating most of her time to navigating, led to marital crisis ending in divorce. Her ex-husband's subsequent unwillingness to provide any financial support, combined with the fact they had previously re-mortgaged the house to finance intervention, led Dilin to financial crisis in the form of personal bankruptcy, where putting food on the table for her three children became a greater concern than her son's autism. The stress of these events, combined with the underlying autism-related stresses and the associated self-neglect, finally took a toll on Dilin's physical health. She developed hypertension and was even hospitalized for a "mild heart attack" while home alone caring for her children. This experience of physical crisis, combined with her doctor's warnings that she "had to find ways to focus on [her]self," forced her to recognize the catastrophic possibility that her children could potentially be left with nothing if she continued at the same pace without tending to her own health. This was the breaking point that pushed her to come to peace with her limits and seek balance both physically and emotionally with a new psychological strategy. "I gave myself permission basically to take some time off," she said, and further reflected that, "one of the things I *have learned* since then is that you need to change your expectation level when you're dealing with autism." Other

caregivers also described a desperate need for "change in direction" that they achieved through psychological strategies or by changing their level of functioning.

While less common in the data, some crises were caused by single acute events. In one example, a caregiver described a single mother she knew who had an autistic young adult daughter. She was running an errand, with her daughter briefly alone at home, when she had an episode of atrial fibrillation complicated by a seriously low heart rate for which she later required a pacemaker. When the ambulance arrived, she explained she could not go to hospital because her daughter was at home and could not care for herself for long periods. After signing a waiver with the ambulance attendant, she drove home despite the strain of her low heart rate, where she called the local autism service agency to arrange for emergency respite before finally admitting herself to the emergency department. In this case, respite relieved the parent's potentially fatal medical crisis.

## Categories of stress

The model of four main categories (i.e., causes or sources) of caregiver stress, and their consequences for caregiver action over time is depicted in Fig 1. *Causes* (or sources) of stress could be distinguished from *consequences* of stress only by analytic reference to the specific empirical instance of stress being described. Thus, common causes of stress (e.g., sleep deprivation, marital problems), could often alternatively be defined as consequences depending on the specific case of stress emphasized in the caregiver's narrative. There were many stories where stress had a feed-forward influence—consequences subsequently causing more stress. The model was developed and verified by tracing numerous paths from causes to consequences, and relating each empirical instance to the stage the caregiver was at in their journey (or process) navigating autism intervention. Each category of stressor is defined and linked to general types of consequences for action or emotional well-being, below. Table 1 shows example quotes and other data from the source grounded theory study that highlight each category.

**Table 1. Four categories of stressors with example data from the grounded theory study.**

| Category | Example data from the grounded theory |
|---|---|
| 1) Sources of urgency to take action | "Well, all of the articles about early intervention—that scared the crap out of me. If your child doesn't get therapy in the first couple of years that they're diagnosed, their prognosis in life or their outcome in life is not reaching their full potential. That scared the crap out of me." (mother of a 7-year-old autistic son, originally diagnosed with 'moderate' autism, and one non-autistic sibling) |
| 2) Buildup of multiple interacting sources | "Another stressor was the space. We had a two-bedroom [making her son's wakefulness at night a particular problem, especially for his sibling]. And me stuck in that 24–7 wasn't helpful." (mother of a 5-year-old autistic son, originally described as 'moderate-to-severe', and a non-autistic sibling) |
| 3) Obstacles causing feelings of helplessness | Regarding the financial barriers to speech and other therapy one mother faced, she commented: "It's really a very stressful feeling, to be honest with you. You feel like you're helpless. You cannot do much. You love your child, and you want to make the sacrifice. And you want to see your child functioning normally. The future—and I've felt worried—I've thought, 'My God, if I die, what will happen to my son? Who will do all this [since] he can't function normally?' It's really hard. It's very, very tough. Every day that passes I think about that." (single mother of a single 6-year-old autistic child, whose needs she described as moderate-to-severe) |
| 4) Depleted resources reducing capacity to cope | "Eventually everything that's in high gear has to stop, or come down, or put the parking brake on for a while, or else it's just going to explode. And that's what started happening. It started taking a toll on me mentally, physically, emotionally—like, everything. Because I wasn't sleeping properly. I wasn't eating properly." (divorced mother of one 6.5-year-old autistic son, originally diagnosed as 'severe,' and two non-autistic siblings) |

1. ***Sources of urgency to take action*** included the following: a) feelings of anxiety about possible future outcomes of autism (e.g., while coming to understand the child has autism [40], or when encountering significant obstacles to intervention), b) the experience of seeing one's child struggle, c) knowing the importance of intervening early, d) learning it's up to caregivers to take action, and e) guilt for not doing enough.

   While still considered distressing and worrisome by caregivers and linked to varying levels of strain on their mental health, these *sources of urgency* generally increased their motivation and capacity to engage in navigating intervention. Such stressors also had the potential to drive caregivers to use personal expendable resources at a rate that was not sustainable in the long term. While the depleted expendable resources this caused down the line represented another stressor (see the fourth category), most participants reported not regretting their losses.

   The timeline and intensity of this category of stress can be mapped onto the trajectory of changes in readiness or motivation for involvement in autism-related care, characterized in detail previously [7]. Namely, it corresponds to the early stage of *going into high gear*, the rapid (steep) increase in motivation for involvement, which occurred for most parents over a period of months starting soon after first noticing developmental concerns in their child. It contrasts, meanwhile, with the subsequent period of *easing off*, defined by a slow (gradual) decline in motivation for involvement generally stretching over several years.

2. ***Buildup of multiple interacting sources*** refers to situations in which stress attributable to the often-overwhelming demands of navigating intervention coincided in time with other stressors such as problematic child behaviors, demands of another newborn, needs of typical siblings, death in the family, or physical aspects of caregivers' daily living environment. It was important to consider how such contextual and potentially difficult-to-identify sources could combine to exacerbate the negative consequences of caregivers' navigation-related stress.

3. ***Obstacles causing feelings of helplessness*** refers to caregivers' encounters with setbacks to their efforts to obtain intervention, which they described as causing a sense of helplessness (loss of control), despair (emotional loss), or reduced capacity to cope emotionally. Obstacles include system-imposed limitations such as inadequate supply of funded intervention resulting in waitlists and qualifying criteria that excluded certain children; and individual actors within the system (professionals, agencies, schools) blocking access to such solutions through discretionary decisions, and sometimes involving interpersonal confrontation, conflict, and power imbalance—contributing stress attributable to the *process* of negotiating obstacles. This resulted in delaying, diminishing or fully denying autism-related care or accommodations sought by the caregiver, which in turn threatened the child's current or future well-being—causing further stress. Additionally, obstacles often worsened caregivers' burden in terms of the scarce expendable resources already invested in pursuing the solution being denied, the extra work of pursuing subsequent alternatives and workarounds, or of otherwise coping with the fallout of unmet needs.

   These obstacles to obtaining care or supports were arguably the most needless and unhelpful sources of stress, especially given the mental anguish they caused. Consequently, some caregivers and professionals highlighted them as priority targets for policy change.

4. ***Depleted resources reducing capacity to cope*** refers to the progressing deficits in caregivers' expendable resources as they were used to meet the work-related demands of autism (parenting and navigating intervention). These sources of stress were analogous to exhaustion because they compromised caregivers' capacity to handle further challenges. Caregivers

often responded by shutting down further resource use, or seeking balance.

This form of stress could have potentially serious consequences by feeding forward to fuel new forms of stress, with one or more possible breakdowns in function (*crisis*), and further reduced capacity to cope or function.

Collectively, the harmful *consequences* of stress included becoming overwhelmed (manifesting as denial, avoidance), self-neglect (disrupted sleep, eating, self-care), onset or aggravation of depression, anxiety, obsessiveness, neglect of family members, guilt, family and marital dysfunction (anger, blame, conflict), divorce or separation, and financial distress.

## Survey data coding and framework analysis exercise

Results of the survey data coding and framework analysis exercise reported here include observations regarding replicability of the grounded theory coding structure, how this coding structure was extended by new open codes from the survey data, and methodological limitations and learnings from applying this approach.

## Replicability of grounded theory coding

S3 Appendix shows the stress-relevant codes that the survey data were coded to—organized within (i.e., as smaller sub-categories of) the four major stress categories described above, and the two categories, *consequences* and *crisis*. They include both original codes from the grounded theory study, and new open codes generated to account for novel ideas recognized after engaging with the survey data. Table 2 summarizes the number of codes (or sub-categories) applied to the survey data within each of these six major categories, and number of references from the survey data to those codes—providing a rough indication of the extent to which the major grounded theory categories were replicated by the survey data. (Note, the more extensive grounded theory study included numerous additional codes not used to code the survey data). Table 2 also indicates the number or new open codes generated from the survey data, and the number of references to them—providing an indication of which categories were extended by the new survey data.

Each major category had some coverage (indicated by number of codes) and level of coding (indicated by number references coded) from the survey data. The first stress category, *sources of urgency to take action* (the largest in the original grounded theory study), had the lowest coverage and level of coding from the survey data. Being action-oriented, this category was more amenable to development in the grounded theory study, which was heavily focused analytically

**Table 2. Summary of survey data coding to six major stress-related categories from the grounded theory study.**

| Category | Number of codes within category (number of new codes) | Number of references to codes within category (number of references to new codes) |
|---|---|---|
| 1) Sources of urgency to take action | 7 (0) | 96 (0) |
| 2) Buildup of multiple interacting sources | 19 (7) | 373 (180) |
| 3) Obstacles causing feelings of helplessness | 43 (1) | 424 (2) |
| 4) Depleted resources reducing capacity to cope | 17 (2) | 270 (9) |
| Consequences | 20 (2) | 137 (3) |
| Crisis | 8 (0) | 41 (0) |

on action and process [7], and therefore dependent on accessing participants' unfolding stories to a deeper level than is practical in surveys with brief open-ended responses.

The third stress category, *obstacles causing feelings of helplessness*, had the highest coverage (number of codes) and level of coding (references). Survey questions about stressors affecting mental health, frustrations with the education system, and reasons why their child had not received adequate support, were the greatest contributors to this category (S2 Appendix). The available data yielded high levels of coding to the specific subcategories (codes) of feeling loss of control, complexity as a setback to access, waitlists, experiencing conflict and lacking expertise within the education system, and a lack of educator capacity to keep the child in school (see S3 Appendix).

The category, *crisis*, was relatively sparsely represented in the survey data, but nevertheless had references coded to all seven of the functioning systems described in the grounded theory-derived definition described above (S3 Appendix). *Consequences* of stress had broad coverage in the sense that a wide variety of effects were represented in the survey data (S3 Appendix), especially related to physical and mental health (for which there were specific survey questions), and self-care.

Survey data coded to the above categories—i.e., excepting the second and fourth stress categories—generally contributed few or no new codes or references to new codes. Results for these two categories are described next.

## Extending the grounded theory coding structure with new survey data-derived codes

The second stress category, *buildup of multiple interacting sources*, was noteworthy for being the most extended by the survey data, with seven new highly referenced codes (S3 Appendix)—the highest number of new codes. All seven reflected the same novel analytic insight: that any family members with special conditions requiring the caregiver's attention (including the caregiver's children, parent(s), spouse, or self) represented a condition contributing to the buildup of multiple interacting sources because they reduced expendable caregiver resources and capacity to respond to stress. This insight was also supported by the grounded theory data. For example, one of the new codes, *multiple autistic children*, was also true for four (13%) caregivers interviewed in the grounded theory study. The survey questions that were the greatest source of coding for this category asked about the effects of stressors on caregiver mental health, and about the contributions of family mental and physical health problems to stress (S2 Appendix).

The fourth stress category, *depleted resources reducing capacity to cope*, was only slightly extended by survey data coding. The survey question producing most data coded to this category asked how stressors impacted caregiver physical health (S2 Appendix). New codes related to time burdens attributable to being a single parent, the lack of capacity for seeing a personal doctor for self-care needs, and effects of physical health problems on mental health and psychological capacity. Additionally, there were extensive references to existing codes of financial and emotional burdens, and to needing a break due to exhaustion.

## Discussion

We have presented the primary conceptual analysis of stress and crisis affecting caregivers of autistic children developed from empirical grounded theory study data, and the results of a subsequent framework analysis exercise to verify and extend this analysis using qualitative survey data. The definitions and conceptual typology resulting from the primary conceptual analysis represents the first dedicated account of stress and crisis we are aware of that is specific to

the experience of caregivers of autistic children. The analysis highlights what makes these caregivers' experience unique by specifically accounting for the causes and consequences of stress as they evolve over their long-term process of navigating care and supports for their child. Being empirically-derived from caregivers' perspectives, its insights are relevant for any professional who supports the stress-related needs of this group. This analysis was derived from data collected in a Western setting (Ontario, Canada), and therefore needs verification in other jurisdictions where variations in systems of care and other contextual aspects may influence the mechanisms of caregiver stress differently. Below, we discuss implications of our new crisis definition for measurement, how our analysis compares to prior stress theory, and methodological learnings and limitations of the framework analysis exercise.

## Crisis definition

We developed the caregiver-centered definition of crisis to expand on and update the only prior published definition of crisis we were aware of that is autism-specific [46, as cited in Weiss & Lunsky, 2011]. Our definition features two important new elements with implications for evaluating crisis in this population. First, it explicitly acknowledges that different systems can be in crisis from the caregiver's perspective (the self, marital unit, family, child, or caregiver-child unit). Second, it explicitly recognizes that crisis can be triggered by progressive buildups of stress, and not just acute (major) triggering events. We are unaware of measurement tools that currently consider these caregiver-relevant aspects. Addressing this measurement gap could improve research understanding and facilitate selecting appropriate targets for intervention when supporting caregivers currently in, or at risk of, crisis, with potential downstream benefits for caregiver and family well-being.

## Comparison to prior stress theory

There are several reasons to compare the conceptual analysis of stress developed from grounded theory data with prior formal stress theories. We followed grounded theory methods [36, 37, 43] by not expanding our knowledge of prior theory until after the study to promote analytic openness to new concepts from the empirical setting. Except for the use of Seyle's General Adaptation Syndrome [45, 53], knowledge of prior formal stress theory was low during the analysis, and did not explicitly influence the resulting definitions and typology. While this theoretically agnostic approach is a strength for developing grounded theory, it may be considered a limitation from other perspectives if the results do not fit with accepted propositions from formal theory. Comparison to prior formal stress theory is therefore important to appraising our analysis and informing decisions about its use. The comparison we provide here includes a consideration of 1) elements from prior theory that are unaccounted for by our analysis, to highlight key aspects from prior stress theory that are useful to consider in addition to what we have presented, 2) consistencies with elements of prior theory, to show the elements from our analysis that reinforce prior theory and vice versa, and 3) novel differences of our analysis, to suggest possible new insights that could extend prior theory.

**What prior theory adds.**   Before comparing prior formal stress theories to highlight the elements from prior stress theory our analysis lacks, it is important to note that the intent of our analysis was never to develop a full theory of stress including its mechanisms. (The objective of the original grounded theory study was to develop a theory of caregivers' process of navigating autism-related care.) Rather, we only defined and conceptually characterized *causes* (sources) and *consequences* (consequences) of stress in the specific population of caregivers of autistic children navigating care. Thus, we did not explicitly account for the third construct common to all true stress theories, *mediators*, necessary to better understand its theoretical

mechanisms [54, 55]. As a minor exception to this, our analysis considers a small number of *coping strategies* as responses to exhaustion, including shutting down further resource use, and seeking balance.

In prior stress theories, these mediators are commonly described to feature *resources* that influence the outcome of stress. Importantly, these resources have been defined more broadly than the *expendable resources* featured in our analysis (i.e., time, energy, and money). In Patterson's Family Adjustment and Adaptation Response (FAAR) model, for example, *resources* are considered the *capabilities* that the family has, in addition to their coping behaviors, to meet the demands that can cause stress [54]. Patterson describes a broad array of personal-, family-, and community-sourced resources, many of which are characterized in other formal stress theories and research. For example, the personal resources, *sense of mastery* and *self-esteem*, and the community resource, *social support*, emphasized in the FAAR model, are also important resources in Hobfoll's Conservation of Resources model [56] and Pearlin's Stress Process model [55, 57]. Similar to FAAR, in those models *resources* underlie stress resistance. But resources in Hobfoll's model are defined more broadly to include things that are either centrally valued in their own right (e.g., self-esteem, close attachments, health, and inner peace) or are a means to obtaining centrally valued ends (e.g., money, social support, and credit) [56]. In a later review, Hobfoll [58] highlighted additional resources, including the key personality-based resources of *self-efficacy*, *dispositional optimism*, and *degree of goal pursuit*, characterizing their relationship to psychological and physical outcomes of stress. Interestingly, Hobfoll originally included a category of resources, *energies*, that is comparable to our expendable resources, which includes time and money that help in the acquisition of other resources [56]. However, all types of resources can be depleted by stressors (external threats), and Hobfoll emphasizes that stress occurs only when there is the loss (or threat of loss) of resources—such as loss of status, position, economic stability, and loved ones. These lost resources represent lost value not only instrumentally, but also symbolically to the extent that they help people define themselves [56]. We note that these and other expanded notions of resources from formal stress theories are coherent with our awareness of the empirical situation of caregivers navigating autism-related care and the stresses they experience. These conceptions of resources therefore have important explanatory power, not contained in our analysis, that should be attended to by anyone needing to understand stress and stress resistance in this population, including intervention and support providers.

**Consistencies with prior theory.** One comparable idea contained in some prior theories is the recognition that stressors (causes) include not only discrete *events*, but also chronic or repeated *strains*. Patterson [54] described strains as conditions of felt tension that usually lack a discrete onset. Pearlin [57], meanwhile, suggested the trajectory of strains can sometimes feature episodic events that embody the same chronic problem, which can cause confusion and delay in defining the underlying strain. While we do not explicitly distinguish strains as a separate class of stressors, we present stress as being caused both by events and by progressive or enduring conditions and meanings. Three categories in our analysis can be considered to primarily comprise strains: *sources of urgency to take action*, which refers exclusively to caregiver-constructed meanings; *buildup of multiple interacting sources*, where the examples presented all represent contextual conditions; and *depleted resources*, which refers to a progressive type of change that is stressful. Patterson's [54] concept of *pileup of demands* refers to the contextual condition of the multiple demands confronting a family at any point in time, which is positioned as useful to understand families' differing responses to a single stressor at a given point in time. Related to this, Pearlin [55] has proposed considering the wider context of life strains as an evolving condition that provides essential explanation for individual responses to more discrete stressors. A prominent type of strain described by both authors is *role strains* [54, 55,

57], which seems highly relevant to explaining stress among caregivers of autistic children. Caregivers' role as navigators of autism-related care is reflected in the label that was given to their problematic social-psychological situation in the original grounded theory study—*[label omitted for double-blind review]*, which is characterized by uncertainty, complexity, lack of support, and a daunting workload [35]. This situation, featuring involuntary role strain, can itself be understood as a major source of stress and a condition predisposing families to crisis. Pearlin et al. [57, 59, 60] have similarly recognized the unexpectedness, high workload, and difficulty of adjustment to emergent caregiving roles, and highlighted how this role itself explains chronic stress itself and downstream stress proliferation (stressors producing further stressors). Our specific analysis here and the broader study from which it was derived [7, 40] provide a further detailed account of how stress related to the role of autism-related care navigation manifests over the long term, reinforcing the importance of understanding chronic or repeated sources of stress, and addressing an empirical gap previously identified by Patterson [54].

A second consistency involves similarities with Patterson's [54] concept of a *pileup of demands*, introduced above. This concept is most evidently analogous to the stress category from our analysis, *buildup of multiple interacting sources*. But it is also similar to the idea from our definition of crisis that crisis sometimes results from a progressive buildup of stress where the final threshold-breaking event can seem relatively benign. Like us, Patterson [54] even used the analogy of the "straw that breaks the camel's back" to support her description of how pileups create conditions where a major event is not necessarily needed to push the demand load beyond the family's management threshold. This common insight highlights the importance of attending to the smaller sources of stress (which may appear as contextual, or at higher systems levels) as a means to help families see the potential of regaining control or balance in the face of crisis [54].

## Novel ideas our analysis adds

A significant new contribution of our analysis involves its conceptualization of social-psychological stress as an evolving process analogous to the physiological conception of stress contained in Seyle's General Adaptation Syndrome model [45, 53]. This metaphor was adopted because it accounted for observed progressive changes in caregivers' responses to, and effects of, stress over time. Namely, it fit the observations of stress motivating and increasing caregivers' capacity for action early on, and later causing exhaustion and breakdown. This time-varying characteristic is missing from other models discussed, and to our knowledge has not previously been applied to social-psychological conceptualizations of stress. Meanwhile, this novel process-oriented understanding is compatible with at least two prior conceptions of stress as an evolving process. Patterson's FAAR model [54] represents the process of stress by outlining how families respond to demands with available capabilities differentially within two smaller processes, pre-crisis adjustment and post-crisis adaptation. And as discussed, Pearlin's [55] Stress Process model recognizes the longitudinally varying nature of stress both by emphasizing the interaction of two types of stressors defined by duration, *discrete events* and *enduring life strains*, and by its mechanistic path from stressful event to stress outcome along which mediators can intervene. While both models resemble our analysis by having early and late phases whose durations are flexible, the different representations of process cannot be assumed to line up across models. Finally, we suggest that our representation of process has potential implications for the measuring stress. Namely, it provides a structure whereby varying measures can be understood (or designed) to differentially measure the intensity of different phases of the stress process—targeting increased motivation and capacity for action in the short- and medium-term, and exhaustion in the long-term. This would fit with Pearlin's

suggestion to pair the measure of stress to the known source of stress by relying variously on situational (short-term) or global (long-term) assessments [55].

**Contrast with homeostatic models.**   Lastly, we highlight a key difference from a salient element of several prior theories. These theories portray stress in terms of an imbalance between demands (including stressors) and capabilities or resources for handling them [45, 53–55, 61, e.g., 62–65]. Hobfoll [56] has criticized such *homeostatic models* as tautological because perceived demands and capacity are defined only in relation to one another. Although we explicitly define *crisis* in homeostatic terms ("when caregiver stress reaches or passes a threshold level that disrupts homeostasis"), we note that our analysis and definition of *stress* contrasts with homeostatic models. Specifically, per our definition, stress does not require a perceived or real imbalance between demands and capabilities, but rather involves the emotional or psychophysiological experience of and response to a perceived threat, regardless of any imbalance. Furthermore, we have portrayed stress as something that can have bidirectional effects on expendable resources (particularly energy levels), increasing capacity for meeting demands for action (especially role demands) at early stages of the long-lasting stress characterizing caregivers' experience, and only decreasing capacity at later stages when exhaustion and depletion of expendable resources is likely.

## Learnings and limitations of the framework analysis exercise

The framework analysis exercise yielded several methodological observations with implications for future use of survey data to extend prior empirical theory. Importantly, while some of the primary grounded theory analysis was reinforced by the survey data coding, it was unreasonable to expect extensive reproduction of its coding structure for at least two reasons. First, the number of stress-relevant codes that could be applied to the survey data was much smaller than for the original grounded theory. This was for two reasons: a larger grounded theory interview dataset relevant to stress (notwithstanding its smaller participant sample) compared to the survey dataset (with only 1,226 mostly brief codable responses); and the relative inability to access depth, complexity, and process with static survey questioning compared to the live, adaptive, and process-focussed interviewing method used in the grounded theory study. Second, the two studies produced subtly different areas of stress-relevant content, both due to different data-generation methods, and to differences in their areas of inquiry.

One survey question—about stress effects on caregivers' physical (medical) health—stood out for producing relevant data on a topic that had little coverage in the grounded theory study. Analysis of this data led to a new insight that extended the grounded theory: that sometimes-serious physical health problems caused by caregiving-related stress could reduce caregivers' capacity for meeting autism-related care demands, both by directly affecting physical capacity, and through effects on mental health and psychological capacity. This demonstrates the potential value of open survey data for extending conceptual analyses from prior qualitative studies. It also highlights the possibility of increasing utilization of survey data, which commonly go under-used. To promote more widespread availability of qualitative survey data for secondary analysis, we suggest survey researchers consider how to embed ethical processes that allow survey participants to consent to such re-use.

While survey data can be useful to extend conceptual analyses, we note two limitations of this data type. First, open participant-typed responses were noticeably less articulate than for transcribed interviews—due to typos, abbreviations, inattentive writing, varying literacy levels, etc. Unlike interviews, surveys do not allow returning to clarify intended meanings. Second, due to their brevity and typical focus on single timepoints, open survey questions are poorly suited to gathering information needed to understand action as a process that evolves over

time or about relative timing of events needed to understand unfolding narratives. Reflecting this limitation, one respondent articulated frustration with not being able to share more within the survey format about how certain stressors had changed over the course of her journey. Grounded theory interviewing, by contrast, is suited to reflect process and narrative timelines [36, 37]—in this case, the evolving nature of and caregiver responses to stress over their trajectory navigating autism-related care.

## Conclusion

Insights from this conceptual analysis of stress among caregivers of autistic children have implications for providing relevant caregiver-centered support for psychological stress and improving assessment of crisis specifically in this population. While not a full theory of stress, multiple key elements are consistent with prior stress theory. A potentially novel contribution is our conceptualization of social-psychological stress as an evolving process analogous to a physiological model of stress, which accurately fits caregivers' experience. The accompanying framework analysis exercise found some utility of qualitative survey data for extending this theoretical analysis, despite limitations of such data compared to in-depth interviews. This has implications for improving utilization of qualitative survey data, which commonly go under-used.

## Supporting information

**S1 Appendix. Grounded theory study interview guide questions related to stress and crisis.**
(DOCX)

**S2 Appendix. Open-ended survey questions containing stress-relevant data, number of codable responses available from each, and observations about the grounded theory categories that the coded data from each question helped replicate most.**
(DOCX)

**S3 Appendix. Summary of stress-related coding of survey data, organized under the four major grounded theory study-derived categories of stress.**
(DOCX)

## Acknowledgments

All authors are grateful both to the parents and professionals who contributed to the primary grounded theory study, and the parents who responded to the survey featured in this report. Stephen Gentles is grateful to his doctoral thesis supervisor, Dr. K. Ann McKibbon, and thesis committee members Drs. Susan M. Jack, David B. Nicholas, and Peter Szatmari, for their supervisory support in completing the grounded theory study.

## Author Contributions

**Conceptualization:** Stephen James Gentles.

**Data curation:** Stephen James Gentles, Janet McLaughlin, Margaret A. Schneider.

**Formal analysis:** Stephen James Gentles.

**Funding acquisition:** Stephen James Gentles, Janet McLaughlin.

**Investigation:** Stephen James Gentles.

**Methodology:** Stephen James Gentles.

**Supervision:** Janet McLaughlin, Margaret A. Schneider.

**Writing – original draft:** Stephen James Gentles.

**Writing – review & editing:** Stephen James Gentles, Janet McLaughlin, Margaret A. Schneider.

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
