## [Decision Letter · Decision Letter 0]

1 Jul 2024

PONE-D-24-18422Stress among caregivers of autistic children: Conceptual analysis and verification using two qualitative datasetsPLOS ONE

Dear Dr. Gentles,

Thank you for submitting your manuscript to PLOS ONE. After careful consideration, we feel that it has merit but does not fully meet PLOS ONE’s publication criteria as it currently stands. Therefore, we invite you to submit a revised version of the manuscript that addresses the points raised during the review process.

We look forward to receiving your revised manuscript.

Kind regards,

Stefaan Six, Ph.D.

Academic Editor

PLOS ONE

Reviewers' comments:

Reviewer's Responses to Questions

**Comments to the Author**

1. Is the manuscript technically sound, and do the data support the conclusions?

Reviewer #1: Yes

Reviewer #2: Yes

2. Has the statistical analysis been performed appropriately and rigorously? 

Reviewer #1: N/A

Reviewer #2: Yes

3. Have the authors made all data underlying the findings in their manuscript fully available?

Reviewer #1: No

Reviewer #2: Yes

4. Is the manuscript presented in an intelligible fashion and written in standard English?

Reviewer #1: Yes

Reviewer #2: Yes

5. Review Comments to the Author

Reviewer #1: This manuscript is well written and easy to follow. The topic of this study is very important. The qualitative analyses process used was well done. The authors did a nice job of highlighting what this study adds or is novel about the findings in this study as well as the limitations of this study.

Reviewer #2: This paper describes a grounded theory study on models of caregiver stress in autism. It is an interesting article and very thoroughly described, however I feel it would benefit from being more concise, therefore I would encourage the authors to try to avoid any repetition and more concisely summarise the study.

Some specific comments are below.

Introduction:

Lines 92-94 seem to repeat lines 69-72.

Methods:

Was the survey data of sufficient depth? Potential bias in terms of some caregivers providing a lot of detail and others giving one word answers, therefore only some responses can be used for qualitative analysis?

The methods might benefit from some additional sub-headings

Results:

Lines 309-318: Some of the findings seem to be caregivers discussing another caregiver – was the study aiming to capture third-hand data such as this and can it be evaluated in the same way as a participant discussing their own experience?

If ethics allows, it would be useful to include some information with quotes, such as the child’s age and severity of autism to identify whether the quotes represent a range from the sample.

6. PLOS authors have the option to publish the peer review history of their article (what does this mean?). If published, this will include your full peer review and any attached files.

Reviewer #1: No

Reviewer #2: No

---

## [Author Response · Author response to Decision Letter 0]

24 Jul 2024

[same as Response to Reviewers file attached]

Response to Reviewers Letter

PONE-D-24-18422

Stress among caregivers of autistic children: Conceptual analysis and verification using two qualitative datasets

PLOS ONE

2024-Jul-05

EDITOR COMMENTS:

1. Please ensure that your manuscript meets PLOS ONE's style requirements, including those for file naming. The PLOS ONE style templates can be found at https://journals.plos.org/plosone/s/file?id=wjVg/PLOSOne_formatting_sample_main_body.pdf and https://journals.plos.org/plosone/s/file?id=ba62/PLOSOne_formatting_sample_title_authors_affiliations.pdf.

AUTHOR RESPONSE: 

We have taken care to follow style requirements as outlined in the templates in the initial submission. Please advise if further changes needed.

AUTHOR RESPONSE: 

Of the multiple funding sources described in the Financial Disclosure, only one has a grant number. We have added this number to the following revised Financial Disclosure statement, and replaced names with initials:

“SJG received a grant from Autism Ontario to support development of this manuscript (grant number N/A). SJG completed the survey data analysis during a postdoctoral fellowship at Wilfrid Laurier University, funded by a gift to JM and the Laurier Autism Research Consortium from the Dare Family, in honor of the late Carl M. Dare (grant number N/A). SJG completed the primary grounded theory study during his doctorate at McMaster University, with the support of two Strategic Training Initiative in Health Research fellowships funded by the Canadian Institutes of Health Research—an Autism Research Training Fellowship jointly funded by the Sinneave Family Foundation, and a Knowledge Translation Canada Fellowship (grant: CIHR 2777601301). JM received an internal grant from Wilfrid Laurier University to support the survey (grant number N/A). The funders had no role in study design, data collection and analysis, decision to publish, or preparation of the manuscript.”

We have added the additional missing funding sources to the Funding Information section.

AUTHOR RESPONSE: 

We have updated our statement on data sharing as below. 

From (OLD statement): Data cannot be shared publicly because of ethical privacy and confidentiality requirements related to the qualitative interview data.

To (NEW statement): Data cannot be shared publicly because of ethical privacy and confidentiality requirements related to the grounded theory study’s qualitative interview data (as enforced by the Hamilton Integrated Research Ethics Board (HiREB)), and the survey study’s qualitative survey data (as enforced by the Wilfrid Laurier University Research Ethics Board). The enforcement by both REBs is governed by Canada’s Tri-Council Policy Statement: Ethical Conduct for Research Involving Humans – TCPS 2 (https://ethics.gc.ca/eng/tcps2-eptc2_2022_chapter5-chapitre5.html#2), and corresponds to the required legal and ethical assurances provided to participants in the consent process about how their privacy and the confidentiality of their data would be protected. Grounded theory or survey data are available for researchers who meet the criteria for access to such confidential data after first contacting either the HiREB (fspence@mcmaster.ca) or Wilfrid Laurier University REB (REB@wlu.ca), respectively.

AUTHOR RESPONSE: This information was in the Methods section of the first submission of the manuscript, but it was embedded in the respective descriptions of the two component studies, and not in a separate “Ethics statement”. We have now moved this information under a new “Ethics statement” subhead at the end of the Methods section. 

AUTHOR RESPONSE: To the best of our knowledge the reference list is complete and correct and does not include any retracted papers. 

REVIEWER #1 COMMENTS:

No comments requesting changes.

REVIEWER #2 COMMENTS:

1. Introduction: Lines 92-94 seem to repeat lines 69-72.

AUTHOR RESPONSE: While the two sentences mentioned reference a similar concept (expendable resource loss), we respectfully suggest they make separate points that are needed to provide important background in the Introduction, and are not redundant. The earlier sentence (lines 69-72), “Combined, these tasks require considerable work, rapid development of expertise (knowledge, skills), and expendable personal resources (time, energy and financial resources) from the caregiver,” describes the burden of the tasks of navigating intervention including the loss of expendable resources. The purpose of the later sentence (lines 92-94), “In addition to being a primary cause of stress, however, care navigation has potentially measurable impacts on personal and family time, energy, and financial resources,” is to highlight how downstream impacts of care navigation (i.e, the three types of expendable resource loss) represent potentially measurable aspects of caregiver stress attributable to care navigation.

2. Methods: Was the survey data of sufficient depth? Potential bias in terms of some caregivers providing a lot of detail and others giving one word answers, therefore only some responses can be used for qualitative analysis?

AUTHOR RESPONSE: Indeed, as we describe in the methods, there was variation in the level of data provided in response to the survey questions that we chose to analyze as containing potentially relevant data (lines 209-215): “Unlike questions in the grounded theory interview guide, these asked less directly about stress or crisis, and several questions addressed service-related aspects expected to be associated with stress. Codable responses ranged from several words to several lines of participant-typed text.” We feel, however, that we have acknowledged and discussed the limitations of this data as a key topic in the Discussion: “While survey data can be useful to extend conceptual analyses, we note two limitations of this data type. First, open participant-typed responses were noticeably less articulate than for transcribed interviews—due to typos, abbreviations, inattentive writing, varying literacy levels, etc. Unlike interviews, surveys do not allow returning to clarify intended meanings. Second, due to their brevity and typical focus on single timepoints, open survey questions are poorly suited to gathering information needed to understand action as a process that evolves over time or about relative timing of events needed to understand unfolding narratives. Reflecting this limitation, one respondent articulated frustration with not being able to share more within the survey format about how certain stressors had changed over the course of her journey. Grounded theory interviewing, by contrast, is suited to reflect process and narrative timelines [36, 37]—in this case, the evolving nature of and caregiver responses to stress over their trajectory navigating autism-related care.” We respectfully suggest that this sufficiently describes the types of “bias” (limitations) embodied in the survey data, specifically with respect to the analytic objectives, and that no additional description is necessary.

3. The methods might benefit from some additional sub-headings

AUTHOR RESPONSE: We have followed this advice by adding the following sub-headings: We regrouped content from both studies related to ethics under a new subheading “Ethics statement.” Additionally, we reworded the two second-level headings to reference the study data being used (to “Primary conceptual analysis of grounded theory data,” and “Conceptual verification and extension using survey data”), and added third-level headings under each.

4. Results: Lines 309-318: Some of the findings seem to be caregivers discussing another caregiver – was the study aiming to capture third-hand data such as this and can it be evaluated in the same way as a participant discussing their own experience?

AUTHOR RESPONSE: In the study, we accepted secondary reports (i.e., other informants’ observations of parents) of caregivers navigating care as key data in this study, both from the 9 professionals interviewed and from parent participants. Such data were considered analytically useful if they could be used in the development of a category or concept. In the case of the lines cited, this data served (i.e., provided a case example) to develop the concept of crisis attributable to a single event (i.e., a medical emergency). This is consistent with grounded theory methods, where it is acceptable to turn to different sources for data that help in conceptual development (e.g., document data, literature review, observation, and different informants).

5. If ethics allows, it would be useful to include some information with quotes, such as the child’s age and severity of autism to identify whether the quotes represent a range from the sample.

AUTHOR RESPONSE: Thank you for this suggestion. We have added information to accompany quotes that describes the parent participant in terms of child age, autism severity, siblings, marital status in some cases, and urban-rural status in some cases. We note that all quotes are from unique participants except one parent who provided two quotes (line 300, and final quote in Table 1).

---

## [Decision Letter · Decision Letter 1]

7 Oct 2024

Stress among caregivers of autistic children: Conceptual analysis and verification using two qualitative datasets

PONE-D-24-18422R1

Dear Dr. Gentles,

We’re pleased to inform you that your manuscript has been judged scientifically suitable for publication and will be formally accepted for publication once it meets all outstanding technical requirements.

Kind regards,

Avanti Dey, PhD

Staff Editor

PLOS ONE

Additional Editor Comments (optional):

Reviewers' comments:

Reviewer's Responses to Questions

**Comments to the Author**

1. If the authors have adequately addressed your comments raised in a previous round of review and you feel that this manuscript is now acceptable for publication, you may indicate that here to bypass the “Comments to the Author” section, enter your conflict of interest statement in the “Confidential to Editor” section, and submit your "Accept" recommendation.

Reviewer #1: All comments have been addressed

Reviewer #2: All comments have been addressed

2. Is the manuscript technically sound, and do the data support the conclusions?

Reviewer #1: Yes

Reviewer #2: Yes

3. Has the statistical analysis been performed appropriately and rigorously? 

Reviewer #1: Yes

Reviewer #2: Yes

4. Have the authors made all data underlying the findings in their manuscript fully available?

Reviewer #1: Yes

Reviewer #2: No

5. Is the manuscript presented in an intelligible fashion and written in standard English?

Reviewer #1: Yes

Reviewer #2: Yes

6. Review Comments to the Author

Reviewer #1: All reviewer comments were responded to and appropriate edits were made. I enjoyed reading this manuscript.

Reviewer #2: (No Response)

7. PLOS authors have the option to publish the peer review history of their article (what does this mean?). If published, this will include your full peer review and any attached files.

Reviewer #1: No

Reviewer #2: No

---

## [Editor Report · Acceptance letter]

11 Oct 2024

PONE-D-24-18422R1 

PLOS ONE

Dear Dr. Gentles, 

I'm pleased to inform you that your manuscript has been deemed suitable for publication in PLOS ONE. Congratulations! Your manuscript is now being handed over to our production team.

Kind regards, 

on behalf of

Dr. Avanti Dey 

Staff Editor

PLOS ONE